# Combined Effects of Citrulline Plus Nitrate-Rich Beetroot Extract Co-Supplementation on Maximal and Endurance-Strength and Aerobic Power in Trained Male Triathletes: A Randomized Double-Blind, Placebo-Controlled Trial

**DOI:** 10.3390/nu14010040

**Published:** 2021-12-23

**Authors:** José Burgos, Aitor Viribay, Diego Fernández-Lázaro, Julio Calleja-González, Josefa González-Santos, Juan Mielgo-Ayuso

**Affiliations:** 1Department of Nursing and Physiotherapy, University of León, 24071 León, Spain; 2Burgos Nutrition, Physiology, Nutrition and Sport, 26007 Logroño, Spain; joseburgos88@hotmail.com; 3Glut4Science, Physiology, Nutrition and Sport, 01004 Vitoria-Gasteiz, Spain; aitor@glut4science.com; 4Department of Cellular Biology, Histology and Pharmacology, Faculty of Health Sciences, University of Valladolid, Campus of Soria, 42003 Soria, Spain; diego.fernandez.lazaro@uva.es; 5Neurobiology Research Group, Faculty of Medicine, University of Valladolid, 47005 Valladolid, Spain; 6Department of Physical Education and Sport, Faculty of Education and Sport, University of the Basque Country, 01007 Vitoria, Spain; julio.calleja.gonzalez@gmail.com; 7Department of Health Sciences, Faculty of Health Sciences, University of Burgos, 09001 Burgos, Spain; mjgonzalez@ubu.es

**Keywords:** citrulline, nitrate, sport performance, beetroot, recovery, triathlon

## Abstract

Citrulline (CIT) and nitrate-rich beetroot extract (BR) are ergogenic aids and nitric oxide (NO) precursors. In addition, both supplements seem to have other actions at the level of muscle metabolism that can benefit strength and aerobic power performance. Both supplements have been studied in numerous investigations in isolation. However, scientific evidence combining both supplements is scarce, and to the best of the authors’ knowledge, there is no current study of endurance athletes. Therefore, the main purpose of this study was to determine the effect of 9 weeks of CIT plus BR supplementation on maximal and endurance-strength performance and aerobic power in male triathletes. This study was a randomized double-blind, placebo-controlled trial where participants (*n* = 32) were randomized into four different groups: placebo group (PLG; *n* = 8), CIT plus BR group (CIT- BRG; 3 g/kg/day of CIT plus 3 mg/kg/day of nitrates (NO_3_^−^); *n* = 8), CIT group (CITG; 3 g/kg/day; *n* = 8) and BR group (BRG; 3 mg/kg/day of NO_3_^−^; *n* = 8). Before (T1) and after 9 weeks (T2), four physical condition tests were carried out in order to assess sport performance: the horizontal jump test (HJUMP), handgrip dynamometer test, 1-min abdominal tests (1-MAT) and finally, the Cooper test. Although, no significant interactions (time × supplementation groups) were found for the strength tests (*p* > 0.05), the CIT- BRG supplementation presented a trend on HJUMP and 1-MAT tests confirmed by significant increase between two study moments in CIT-BRG. Likewise, CIT-BRG presented significant interactions in the aerobic power test confirmed by this group’s improve estimated VO_2max_ during the study with respect to the other study groups (*p* = 0.002; η^2^p = 0.418). In summary, supplementing with 3 g/day of CIT and 2.1 g/day of BR (300 mg/day of NO_3_^−^) for 9 weeks could increase maximal and endurance strength. Furthermore, when compared to CIT or BR supplementation alone, this combination improved performance in tests related to aerobic power.

## 1. Introduction

The triathlon is an event that combines the disciplines of swimming, cycling and running in this sequential order [1,2]. Although distances vary, all triathlons can be considered as events in which aerobic power predominates, since the physiological parameters associated with endurance training are considered as relevant; maximal oxygen consumption (VO_2max_), anaerobic/ventilatory threshold and economy of movement, among others [3,4]. In addition, strength performance plays a key role since, it is a sport modality in which tension against loads is developed during contraction [5]. The high physical demands of this sport make nutritional recommendations insufficient, so triathletes resorts to nutritional supplements to improve their sport performance [6]. Among these, there is increasing scientific evidence of the benefits that nitric oxide (NO) precursors (L-citrulline (CIT) and supplements rich in nitrates (NO_3_^−^) such as beetroot juice or nitrate-rich beetroot extract (BR) provide for sports performance, since these ergogenic aids could improve both aerobic and anaerobic metabolism [7]. In particular, NO is involved, among other factors, in the regulation of blood flow, muscle contractility and mitochondrial respiration [8]. Concretely, the NO is synthesized by oxidation of the amino acid L-arginine (ARG), which comes from CIT or by reduction of NO_3_^−^ to nitrite [9].

On the one hand, CIT is a non-essential amino acid mainly found in watermelon, but it can also be endogenously produced via synthesis from glutamine and L-arginine (ARG) to the NO conversion pathway [10]. In addition, the ability of CIT to buffer the acidosis, hyperammonemia and blood lactate accumulation has been shown to provide performance benefits in athletes performing training sessions which involve aerobic power or strength performance [11]. Concretely, 6 g/day of CIT supplementation for 7 days appears to increased exercise tolerance and total work on aerobic power performance [12]. Moreover, 2.4 g/day of CIT supplementation increased VO_2_ uptake kinetics during endurance tests after 7 days of supplementation [13]. This dose significantly reduced the time needed to complete a 4 km time trial [13] and improved training volume by increasing fatigue tolerance [14]. On the other hand, it has been shown that CIT can stimulate the mammalian Target of Rapamycin (mTOR) pathway, independent of insulin, the growth hormone (GH) or type 1 insulin-like growth factor (IGF1), improving muscle protein synthesis (MPS) [15]. In this sense, CIT has shown improvements in isometric muscle strength levels [16]. Along the same lines, a single dose of CIT (8 g) supplementation prior to exercise improved the number of bench press repetitions performed at 80% at 1 RM [17]. It also leads to a significant reduction in the sensation of fatigue by increasing the rate of adenosine triphosphate (ATP) production during exercise and the rate of phosphocreatine recovery after exercise [18]. These mechanisms cause CIT to be presented as a potential aid in the improvement of aerobic and strength performance.

On the other hand, athletes source NO_3_^−^ supplements from beetroot, such as beetroot juice or BR, to improve their aerobic performance [19,20]. One possible explanation for this ergogenic effect of NO_3_^−^ may be an increase of NO which increases blood flow in muscle during exercise [21,22]. In this sense, supplementation with 0.5 L of juice 2 h before exercise appears to improve aerobic power in submaximal aerobic tests, optimizing the total time in a trial in trained cyclists [23]. However, 210 mL of juice (19.5 mmol~1.1 g of NO_3_^−^) both for one day (acute) and for 8 days (chronic) before completion of a submaximal treadmill run and 1500 m time trial (aerobic power) did not reduce VO_2_ nor time-trial in a group of elite distance runners [24]. On the other hand, NO_3_^−^ could play a key role in anaerobic exercises, maximal strength or endurance-strength performance [25,26]. Besides, the contractile characteristics of human muscle are improved by dietary NO_3_^−^ [27]. This effect could be attributed to nitrosylation of the ryanodine receptor and enhanced NO signaling through the soluble guanyl cyclase-cyclic guanosine monophosphate-protein kinase G pathway, both of which increase free intracellular Ca^2+^ concentration and myofilament Ca^2+^ sensitivity [28,29]. Therefore, Nyakayiru et al. reported that 140 mL of BR juice supplementation 2.5 h before exercise improved isometric force production [30]. In addition, other studies presented positive results in achieving a greater number of repetitions of bench press by ingesting a drink (400 mg of NO_3_^−^) for 6 days [26]. In this regard, BR supplementation may improve mitochondrial efficiency [31] and/or reduce the cost of ATP in both maximal and endurance-strength performance production [32].

Therefore, these two supplements (CIT and BR) have been extensively studied individually, showing improvements when supplemented acutely (30–60 min before exercise) or in the short-term (7 days to 4 weeks) [33,34,35]. However, a possible combination of both for long-term (9 weeks), could further improve performance than either CIT or BR separately. In this line, our research group reported that 10 weeks of creatine monohydrate with β-hydroxy-β-methyl-butyrate (HMB) oral co-supplementation showed a synergistic effect by increasing athletic performance, decreasing exercise-induced muscle damage and regulating hormonal behaviors in comparison to when they were taken in isolation [36,37]. This could be beneficial for post-exercise metabolic/physiological adjustments. In this regard, to the best of the authors’ knowledge, only one study has analysed the potential benefits of acute CIT plus BR combination in healthy older adults [38]. In this research, the supplementation of 6 g of CIT plus 520 mg of NO_3_^−^ 6 h before a submaximal incremental cycling test showed improvements in some cardiorespiratory variables, such as VO_2_, but did not present a positive effect on knee extension exercise performance [38]. In addition, exercise may affect NO bioavailability due to both impaired NO activity and lack of NO substrate [39].

In this way, it can be hypothesized that the synthesis of NO would be used independently by both routes (BR and CIT). Additionally, long-term supplementation with a smaller dose could keep NO levels elevated for longer periods, optimizing endogenous NO_3_^−^ pools [22,40]. Therefore, the main aim of this study was to determine the effects of long-term (9 weeks) oral co-supplementation with 3 g/day of CIT plus 2.1 g/day of BR (300 mg/day of NO_3_^−^) on maximal strength by maximal horizontal jump (HJUMP), and the handgrip dynamometry test (DYN), strength-endurance performance by 1-min abdominal test (1-MAT) and aerobic power by Cooper’s test in male trained triathletes.

## 2. Materials and Methods

### 2.1. Participants

Thirty-two male amateur triathletes (32.17 ± 4.87 years; 22.57 ± 1.79 kg/m^2^ and 7.8 ± 1.05% of fat mass) with more than 5 years of experience in triathlon performance participated in this randomized double-blind placebo-controlled trial. All triathletes completed 6 training sessions per week during the 9 weeks (period of the study). They all completed the same workload volume (70% aerobic work, 20% of strength sessions in the gym and an additional 10% core, injury prevention or joint mobility drills) with a duration of 2.5 h/training session. The total average hours of exercise during the study were 135 h.

In addition, a registered dietician nutritionist (registration number: CLR-0020) elaborated a personal nutritional plan for each triathlete. This weekly dietary plan was developed to optimize the health and performance of each participant, with the aim of providing the necessary intake as well as the optimal amount of macro and micronutrients according to the training load, considering the personal characteristics and intolerances of each [41]. To control for all athletes following the planned diets, they performed two validated methods of dietary recall [42]. The first method was a food frequency questionnaire (FFQ), previously used in other studies with athletes [43,44], at the end of the study (T2). The second method was a seven-day dietary recall collected the week prior to T1 and during the week of T2. This method was used to check if the results of the FFQ were similar to those of this recall [45].

All athletes also underwent a medical examination and completed a medical history questionnaire prior to the start of the study to find out whether they had any type of disease and/or injury. No participants had any diseases, and they did not smoke, drink alcohol, or take other medications, stimulant substances or drugs which would alter the hormonal responses. Likewise, to avoid the possible interference of other nutritional supplements on the different variables measured in this study, a 2-week washout period was introduced. During the 9 weeks of research, each participant only took the assigned supplement and abstained from any other type of supplement.

All participants were fully informed of all procedures of the study and signed a statement of informed consent. This research was designed in accordance with the Declaration of Helsinki (2008) and the Fortaleza update (2013) and was approved by the Human Research Ethics Committee at the University of León, León, Spain (number: ULE-020-2020). Likewise, this study was registered in clinicaltrials.gov with NCT05143879 number.

### 2.2. Experimental Protocol and Evaluation Plan

This study was designed as a randomized, double-blind and placebo-controlled trial study in order to evaluate the potential effects of the combination 3 g/day of CIT plus 2.1 g/day of BR (300 mg/day of NO_3_^−^) on sports performance during 9 weeks via several tests: HJUMP, DYN, 1-MAT and the Cooper test. The doses and supplementation time proposed for this study were based on studies that had found favourable results with similar doses [33,34,35].

The participants were randomized to four previously described groups by an independent statistician using OxMaR open-source software^®^ (Oxford Minimization and Randomization, 2014): (I) placebo group (PLG): *n* = 8; height: 179 ± 8 cm and body mass: 73.5 ± 5.4 kg; (II) CIT group (CITG): *n* = 8; height: 180 ± 9 cm and body mass: 69.9± 7.8 kg; (III): nitrate-rich beetroot extract group (BRG): *n* = 8; height: 178 ± 8 cm and body mass: 71.6 ± 6.3 kg and (IV) CIT-BR group (CIT- BRG): *n* = 8; height: 181 ± 6 cm and body mass: 73.6 ± 5.8 kg.

The supplementation, including the placebo (cellulose), was taken in 6 capsules/day (3 capsules of 1 g of CIT by Hard Eight Nutrition LLC^®^ (7511 Eastgate Rd, Henderson, NV 89011) and 3 capsules of 700 mg of BR which providing 100 mg of NO_3_^−^ by Lindens Health Nutrition^®^ (1 Calder Point, Monckton Road, Wakefield, WF2 7AL), 2 (breakfast), 2 (lunch) and 2 (dinner) during the 9 weeks of the study. Weekly, a registered dietician (LR003), who had prepared the capsules in both cases (athletes and sports team) did not know their contents (double-blind), gave each athlete their supplement and checked that the protocol was being carried out correctly under the guidelines.

### 2.3. Performance Assessment Tests

All participants attended the laboratory at 8:30 a.m. to complete the performance test at two specific moments during the study: (T1) at baseline and (T2) post-intervention after 9 weeks of supplementation. The dietician nutritionist prepared each triathlete’s personal diet for the 48 h before testing based on a competition recommendation diet [41].

The tests were always supervised by the technical team of the study. The athletes were familiarized with these tests. Before testing, a standardized 15-min warm-up was performed: (I) 8-min incremental run; (II) 3-min of Core work; (III) 2-min exercises for trunk, hip, and leg muscles; and (IV) 2-min different types of jump. After warm-up, the different performance tests were carried out in the sequence below.

#### 2.3.1. Horizontal Jump Test (HJUMP)

The length reached by each participant in each study group was assessed from the take-offline to the point where the rearmost part of the foot, the heel, landed on the surface. The length reached by the horizontal jump was measured by laying a tape measure on the ground with an accuracy of 0.1 meters and precision of 1 cm, analyzing the force-time values and derived by numerical integration, the speed-time values, the product of both variables being the power time values. The final length was obtained from the best of the two jumps [46].

#### 2.3.2. Handgrip Dynamometer Test (DYN)

An isometric grip strength test was performed using a handgrip dynamometer (DYN). Prior to testing, and in order to optimize the results, the grip was adjusted for each participant according to their hand size [47]. The protocol consisted of each athlete, in a bipedal position, gripping the dynamometer with the dominant hand and providing maximum force for 4 s [48]. This protocol was repeated on 3 occasions with 1 m recovery breaks between repetitions, and the highest result among these 3 values (Kg/m⋅s^2^) was used [47].

#### 2.3.3. 1-Min Abdominal Test (1-MAT)

An abdominal test was performed to assess the maximum number of repetitions completed in 1 min. Each repetition was validated from a starting position where the shoulder blades touched the surface of the floor until reaching the top, where both elbows had to touch the knees. The researcher only counted repetitions that followed this protocol [49,50].

#### 2.3.4. Cooper Test

The Cooper test was conducted under the observation of the research team on a 400-m synthetic official sports track. Triathletes were familiar with this test given that they usually use it throughout the season. The participants completed the traditional test protocol, which consisted of covering the farthest distance feasible in 12 min [51]. The total distance covered in this time was measured immediately after the test was completed using markers put on the track at 50-m intervals [52]. Subsequently, estimated VO_2 max._ was calculated using the equation 22.351 × Distance covered (in kilometers)-11.288.

### 2.4. Anthropometry

The International Society for the Advancement of Kinanthropometry (ISAK) protocol was used for anthropometrical measures [45]. In addition, all participants were measured by the same internationally certified anthropometrist ISAK 3 (certificate number #636739292503670742). All measurements were taken twice to establish the reliability of the new test. If the difference between both measures exceeded 5% for an individual skinfold, a third measurement was taken. The mean of the measurement taken twice, or the median of the triplicate anthropometric measurements were used for all the analyses. Height (cm) was measured using a SECA^®^ measuring rod, with 1 mm precision, while body mass (kg) was assessed by a SECA^®^ model scale, with 0.1 kg precision. Body mass index (BMI) was calculated using the equation weight (kg)/height (m^2^). Eight skinfolds (mm) were measured: triceps, subscapular, bicipital, supraspinatus, supraspinal, abdominal, front thigh and medial calf with a Harpenden^®^ skinfold caliper with 0.2 mm precision, and the sum of these 6 skinfolds was calculated. Girths (cm) (relaxed arm, flexed arm, minimum waist, 1 cm below the buttock thigh, mid-thigh and calf) were measured with a narrow inextensible metallic Lufkin^®^ measuring tape model W606PM with a precision of 1 mm. Muscle mass (MM) was estimated by Lee and somatotype components [45].

### 2.5. Statistical Analysis

The study variables are represented as means and standard deviations for T1 and T2. The percentage of change (Δ: %) between T1–T2 was calculated by the following equation: ((T2 − T1)/T1) × 100 for each study group. To analyze the normality of the data for each of the variables, the Shapiro-Wilk model (*n* < 50) was used. Levene’s test was applied to determine the homoscedasticity of the variances. A two-way repeated measure analysis of variance (ANOVA) test was used to examine interaction effects of time with the supplementation in the different performance tests among the different study groups (PLG, CITG, BRG, CIT-BRG). A Bonferroni post-hoc test was subsequently applied for comparisons among the different study groups. Partial eta squared (n^2^p) was used to calculate effect sizes among participants. Since this measure is likely to overestimate the effect sizes, the values were interpreted according to Ferguson [53].

The analyses were performed using SPSS^®^ software version 24.0 (SPSS, Inc., Chicago, IL, USA) and graphics using GraphPad Prism 6 software (GraphPad Software, Inc., San Diego, CA, USA). Statistical significance was accepted when *p* < 0.05.

## 3. Results

The data obtained on body composition and somatotype showed no significant differences (*p* > 0.05) in the group-time interaction (body mass, muscle mass and percentage of fat mass) (Table 1). However, significant increases (*p* < 0.05) were observed between the two time points of the study (T1 vs. T2) for muscle mass (%) and mesomorphy in CIT + BRG (Table 1).

Table 2 shows the values of the different variables analysed during the physical tests performed in both T1 and T2 moments. Significant group-by-time interactions can be observed in the group-by-time interaction in estimated VO_2max_ by Cooper test (*p* = 0.002; ƞ^2^p = 0.418). However, there were no significant differences in the group-by-time interaction for HJUMP, DYN and 1-MAT test (*p* > 0.05). In addition, significant increases (*p* < 0.05) between two study moments were observed on 1-MAT in CIT-BRG (T1: 49.3 ± 9.7 vs. T2: 56.6 ± 11.4 repetitions), HJUMP (T1: 2034.1 ± 114.84 vs. T2: 2114.6 ± 170.1 cm) and in Cooper’s test (T1: 2953.00 ± 372.7 vs. T2: 3079.6 ± 423.5 m). Finally, all groups showed significant increases in DYN after 9 weeks of supplementation (*p* < 0.05).

Figure 1 shows the percentage change in the different physical tests for each of the study groups. Significant differences can be observed for the Cooper test (*p* = 0.003; n^2^*p* = 0.418). Concretely, CIT-BRG presented significantly higher value of % change than PLG and CITG (*p* < 0.05). However, although no significant differences were found for maximal strength (HJUMP and DYN) and resistance-strength (1-MAT) performance test (*p* > 0.05), the CIT-BRG presented higher % change in HJUMP and 1-MAT than other groups.

## 4. Discussion

To the best of the authors’ knowledge, this is the first study that has evaluated the effect of long-term (9 weeks) oral co-supplementation with 3 g/day CIT plus 2.1 g/day BR (300 mg/day of NO_3_^−^) on maximal strength, endurance-strength performance test and aerobic power test in male triathletes. The main results indicated that this combination presented group-by-time in aerobic power measured by the Cooper Test, showing an increase in estimated VO_2max_ with respect to PLG and CITG. On the other hand, although the maximal strength performance by HJUMP and DYN, and endurance-strength performance by 1-MAT, did not show significant differences in the group-time interaction, the CIT-BRG showed a positive trend with respect to the other supplementation groups in the same tests. Moreover, only the CIT-BRG displayed a significant increase on HJUMP and 1-MAT after 9 supplementation weeks.

The use of nutritional supplements to improve sports performance is becoming increasingly widespread among triathletes [54,55]. In this regard, CIT [12,13,14,15,17] and BR (NO precursors) [23,25,26,30,31,32] are two nutritional supplements that could help improve performance in both exercises, involving neuromuscular strength and endurance [56,57]. It has been shown that NO can help athletes with functions that are closely involved in the regulation of muscle contraction, blood flow, maximal oxygen consumption during exercise, adaptations at the mitochondrial level and homeostasis of glucose and calcium [27,57,58,59], which in the long term can improve athletic performance [60]. These NO precursors may also improve performance in maximal and endurance strength and aerobic power tests through an increase in ATP production, improvement in contractile function and a decrease in ammonium, etc. [18]. Furthermore, both supplements appear to have other actions at the muscle metabolism that could help with strength and aerobic power performance.

To the authors’ knowledge, the effects of the long-term combination of CIT plus BR on maximal and endurance-strength performance have not been studied in depth [38]. However, some research that has evaluated the acute effects of these supplements individually has found improvements in strength performance. Mosher et al. [26] found significant differences in the number of bench press repetitions performed at an intensity of 60% of their 1-RM for 3 sets to failure, with 2 min rest interval between sets, comparing the group supplemented with 70 mL of BR and the PLG group. In the same line, Haider and Folland, presented significant differences in the number of repetitions in leg press after 7 days of supplementation with a concentrated BR rich in NO_3_^−^ (70 mL of BR) taken 2.5 h before exercise [61]. These authors indicated that the observed effects were due to an improvement of the contractile properties of the human skeletal muscle. Specifically, NO_3_^−^ supplementation improved the excitation–contraction coupling at low stimulation frequencies improving explosive strength production [61].

On the other hand, and regarding endurance-strength capacity, Perez-Guisado showed a significant increase in the number of bench press repetitions performed and a reduction in fatigue at 24 and 48 h after ingestion of 8 g of CIT one hour before training [62]. Along the same lines, Wax et al. [17] obtained satisfactory results by performing a similar protocol increasing upper-body resistance performance in trained college-age males. Moreover, a recent meta-analysis concludes that CIT could have a slight positive effect on strength performance, but raises many doubts as to the effective doses and mode of use, but indicating that the approach might make sense especially in highly competitive athletes [35]. In the present study, although there were no significant differences among groups in both analysed strength performances, the CIT-BRG presented a better trend with respect to other groups in jump tests and 1-MAT. These results could be confirmed by a significant increase in the CIT-BRG between study moments in these strength tests, reinforcing the original hypothesis in which it was considered that both supplements could act in a complementary way improving strength performance via an improvement in excitation-contraction coupling and preventing fatigue metabolites accumulation [28,61]. However, more studies must be carried out with larger samples to determine if this trend is effective.

In the same way, as strength performance, the effects of the long-term combination of CIT plus BR supplementation on aerobic power have not yet been described. However, some authors have shown the effect of both supplements individually on this capacity. Nyakayiru et al. indicated that the increase in the distance covered in an intermittent test (YO-YO 1R1) was greater in the group supplemented with 140 mL of BR (~800 mg NO_3_^−^/day) for 6 days [30]. Other authors have found similar improvements on 10-km cycling time trial when the cyclists were supplemented with 12.4 mmol NIT/day and ~8 mmol NO_3_^−^/day for 7 days, respectively [19,23]. Regarding CIT supplementation, Suzuki et al. showed a significant increase in the distance covered in a 4 km cycling time trial after 2.4 g of CIT for 7 days [13]. Along the same lines, although at lower intensities than maximum aerobic power, Stanelle et al. performed a 40 km incremental distance test in which they showed a 5.2% reduction in total time compared to the PLG group when 6 g of CIT were administered for 7 days, which may explain similar physiological and metabolic effects [12]. In this case, as happened in the maximal and endurance strength performance, the combination of both supplements showed that CIT-BRG presented a significant increase after 9 weeks, with respect to other groups, in aerobic power.

These results can be explained by the sum of the same and different ways in which both supplements can be beneficial in sports performance. Among reported physiological effects of these supplements’ intake, authors consider as potential effects that could explain results the better dynamics of VO_2_, greater vasodilation and nutrient delivery to muscle, lower accumulation of fatigue metabolites, higher ATP and oxidation rate of pyruvate and improved contractile function [11]. In addition, this could be reinforced by long-term supplementation with a lower dose, which could maintain the NO_3_^−^ pool in the muscle and the production of ON and prevent ON decline, because exercise may affect NO bioavailability due to both impaired NO activity and lack of NO substrate [63,64].

In this sense, the sensitivity of skeletal muscle to NO_3_^−^ availability and the dynamic changes of these, together with NO_3_^−^ during exercise and subsequent recovery [65], certainly hint at a physiological role that has been underestimated to date, with skeletal muscle being fundamental to the maintenance of NO in the human body [59,66]. In fact, it has been evidenced that skeletal muscle has the capacity to accumulate, transfer and metabolize NO_3_^−^ and nitrite [67]. Therefore, prolonged supplementation with this type of ergogenic aid (BR) would increase the levels of stored NO_3_^−^, favourable for the production of NO and, subsequently, associated with long-term adaptations [68]. In this regard, acute or chronic administration of NO_3_^−^ by athletes has also been demonstrated to improve the contractile properties of human skeletal muscle, particularly the twitch force, rate of force development, estimated maximal shortening velocity and maximal power of muscle [28]. The exact mechanisms responsible for the increase in human muscle contractility produced by NO_3_^−^ are still unknown, but alterations in Ca^2+^ signaling due to increased bioavailability could play a key role [28,29]. All these mechanisms probably act in a complementary manner improving sports performance due to their vasodilator, muscle function and mitochondrial respiration modulating effects, to which the improvement of certain capacities related to sports performance is attributed [69].

In addition to the effect on NO, CIT has been shown to stimulate muscle protein synthesis by activating mTOR through the phosphatidylinositol 3-kinase/mitogen-activated protein kinase/factor 4E-binding protein 1 (PI3K/MAPK/4E-BP1) [70] pathway, and by increasing ARG production, which promotes growth hormone secretion [71]. Likewise, the increase in ARG production will improve intramuscular creatine levels, which will allow an increase in phosphocreatine reserves, being a key amino acid in the urea cycle, and may increase ammonium elimination, leading to improvements in high intensity performance due to the decrease in lactate accumulation and the increase in ATP resynthesis [72], thus reducing accumulated fatigue during exercise [73]. The CIT is a key activator of muscle protein synthesis during periods of high intensity training, through activation of the mTOR pathway due to its key role in the regulation of nitrogen homeostasis, which may play a potential role in recovery and training load assimilation [74,75]. Considering that CIT is co-produced with NO as a product of NOS activity through the CIT-NO cycle, administration of exogenous CIT could represent a potential alternative to increase the amount of ARG supplied to NO. In this sense, the long-term effect of CIT, together with its potential effect on mTOR, could impact on testosterone production by potentiating NO and, therefore, increasing blood flow through the testis [76]. Consequently, it would promote more efficient transportation of O_2_ to the blood during muscle contraction, obtaining a better skeletal muscle response [77]. In addition, CIT mitigates the production of lactate by allowing the production of pyruvate for energy utilization leading to an increase in the efficiency of ATP production, which may be a determinant in high intensity performance [11]. This relationship may explain the increase in distance run shown in this study, CIT being a key amino acid in the urea cycle, thus being able to increase ammonium elimination, which would lead to these improvements due to decreased blood lactate and increased phosphocreatine and ATP resynthesis [72].

### 4.1. Limitations, Strengths and Future Research

It should be noted that it is difficult to obtain larger samples in athletes, as not many have the availability to comply with the training and supplementation instructions required by the study. Moreover, sampling using a convenient, non-probabilistic sampling procedure may produce results that are not representative of the rest of the population. These limitations may underrepresent the results and may affect study outcomes. Nevertheless, the purpose of this study is not to transfer information to the general population.

However, the methodology used in this trial, randomized double-blind, and placebo-controlled is its most important strength. In addition, another strength was the control of the triathletes’ diet, as well as the control of body composition throughout the intervention process, so that these outcomes did not influence the final results based on present data.

Future research should continue the study of the long-term effects of this combination in different tests related to neuromuscular strength, in order to expand existing knowledge about this potential combination. It should also examine the efficacy of these supplements in experienced athletes, to determine whether the use of these supplements as part of treatment could increase athletic performance. In addition, it should analyze how this possible combination affects the female population for anaerobic sports, given that this study only focused on men and their performance.

### 4.2. Practical Applications

This research could be of interest to physicians and nutrition practitioners who want to improve athletes’ performance in aerobic power tests. Considering that 3 g/day of CIT plus 2.1 g/day of BR (300 mg/day of NIT) for 9 weeks period of time could increase the distance covered and, therefore, the VO_2max_, these could be considered as supplementation phases in periods of intensive practice.

## 5. Conclusions

In conclusion, although the combination of 3 g/day of CIT plus 2.1 g/day of BR (300 mg/day of NIT) did not show significant interactions among study groups during study of maximal strength and endurance-strength performance, only the CIT-BRG presented significant improvements in HJUMP and 1-MAT after 9 weeks of supplementation.

In addition, the combination of 3 g/day of CIT plus 2.1 g/day of BR (300 mg/day of NIT) for 9 weeks showed significant interactions in aerobic power performance among groups during the study. These interactions were confirmed with higher estimated VO_2max_ by CIT-BRG with respect to the other study groups.

Therefore, the combined supplementation of 3 g/day of CIT plus 2.1 g/day of BR (300 mg/day of NIT) for 9 weeks could improve maximal strength and endurance strength. Besides, this potential combination promotes an increase in performance (in tests involving aerobic power) with respect to CIT or BR supplementation separately.

## Figures and Tables

**Figure 1 nutrients-14-00040-f001:**
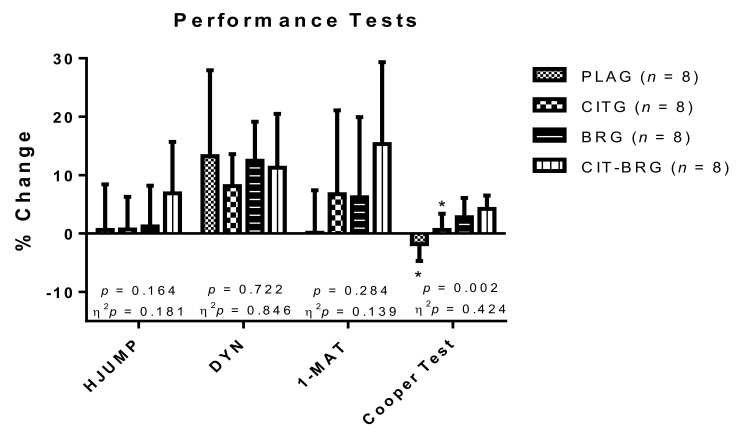
Percentage changes during the study in maximal strength, endurance-strength, and aerobic power in the triathletes. Data are presented as mean ± standard deviation. Δ: ((T2 − T1)/T1) × 100. *: Significant differences with respect to CIT-BRG (*p* < 0.05).

**Table 1 nutrients-14-00040-t001:** Body composition and somatotype data in the four study groups at the baseline (T1) and after 9 weeks of supplementation (T2).

Group	T1	T2	P (T × G)	ƞ^2^*p*
6 Skinfolds (mm)				
PLG	61.10 ± 19.46	57.77 ± 20.37	0.251	0.140
CITG	55.75 ± 13.68	49.28 ± 12.53		
BRG	61.66 ± 19.86	53.15 ± 9.11		
CIT-BRG	46.95 ± 15.82	47.53 ± 20.61		
Muscle mass (%)				
PLG	43.43 ± 3.28	44.47 ± 3.69	0.137	0.195
CITG	48.42 ± 5.85	45.86 ± 3.78		
BRG	45.09 ± 3.11	46.44 ± 3.34		
CIT-BRG	45.56 ± 3.01	48.00 ± 2.10 *		
Endomorphy				
PLG	2.82 ± 1.03	2.65 ± 0.87	0.482	0.089
CITG	2.33 ± 0.72	1.97 ± 0.46		
BRG	2.68 ± 0.44	2.33 ± 0.46		
CIT-BRG	2.24 ± 0.87	2.17 ± 0.84		
Mesomorphy				
PLG	5.20 ± 0.41	5.33 ± 0.60	0.568	0.079
CITG	5.38 ± 0.89	5.86 ± 1.13		
BRG	5.37 ± 0.95	5.66 ± 1.11		
CIT-BRG	4.70 ± 0.70	5.03 ± 0.81 *		
Ectomorphy				
PLG	2.48 ± 0.80	2.44 ± 0.66	0.746	0.031
CITG	2.62 ± 0.84	2.46 ± 0.77		
BRG	2.59 ± 0.92	2.58 ± 0.93		
CIT-BRG	3.03 ± 0.87	3.03 ± 0.83		

Data are expressed as mean ± standard deviation. P (T × G): group-by-time interaction (*p* < 0.05). Two-way repeated-measures ANOVA. PLG: Placebo group; CITG: Citrulline supplemented group; BRG: Nitrate-rich beetroot extract group; CIT-BRG: Citrulline plus Nitrate-rich beetroot extract group. *: Significantly different between two study phases (T1 vs. T2) by dependent sample *t*-test (*p* < 0.05).

**Table 2 nutrients-14-00040-t002:** Maximal strength, endurance-strength, and aerobic power in the four study groups at baseline (T1) and after 9 weeks (T2).

Group	T1	T2	P (T × G)	η^2^*p*
Horizontal jump test (cm)				
PLG	2145.8 ± 163.7	2156.7 ± 219.6	0.696	0.049
CITG	2214.7 ± 221.5	2227.1 ± 214.9		
BRG	2113.8 ± 164.4	2143.9 ± 256.4		
CIT-BRG	2034.1 ± 114.8^.^	2114.6 ± 170.1 *		
Handgrip dynamometer test (Kg/m·s^2^)				
PLG	51.13 ± 4.70	57.13 ± 9.11 *	0.607	0.092
CITG	53.75 ± 4.92	63.00 ± 4.24 *		
BRG	56.33 ± 2.66	60.51± 5.28 *		
CIT-BRG	51.43 ± 6.29	56.71 ± 9.62 *		
1-Minute abdominal test (repetitions)				
PLG	51.1 ± 13.3	50.5 ± 10.2	0.145	0.191
CITG	48.0 ± 7.8	50.7 ± 7.5		
BRG	54.9 ± 8.0	59.1 ± 11.8		
CIT-BRG	49.3 ± 9.7	56.6 ± 11.4 *		
Estimated VO_2max_ (mL/kg/min)				
PLG	58.3 ± 5.5	56.7 ± 5.9	0.002	0.418
CITG	56.6 ± 7.9	5.6.5 ± 8.1		
BRG	57.6 ± 9.3	59.5 ± 9.6		
CIT- BRG	54.7 ± 8.3	57.5 ± 9.5 *		

Data are expressed as mean ± standard deviation. *p* (T × G): group-by-time interaction (*p* < 0.05. all such occurrences) by two-way repeated-measures ANOVA. *: Significantly different between two study phases (T1 vs. T2) by dependent sample t-test (*p* < 0.05). PLG: Placebo group; CITG: Citrulline supplemented group; BRG: Nitrate-rich beetroot extract group; CIT-BRG: Citrulline plus Nitrate-rich beetroot extract group.

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
