# Peer review of "Combined Effects of Citrulline Plus Nitrate-Rich Beetroot Extract Co-Supplementation on Maximal and Endurance-Strength and Aerobic Power in Trained Male Triathletes: A Randomized Double-Blind, Placebo-Controlled Trial"

_nutrients, 2021, doi:10.3390/nu14010040_

Round 1

Reviewer 1 Report

General comment: The research article entitled “Combined effects of citrulline plus nitrate-rich beetroot extract supplementation on maximal and endurance-strength and aerobic power in trained male triathletes: A randomized double-blind, placebo-controlled trial” is a well-organized study, with sufficient methodology and adequate description of the results. Some minor corrections are required for the improvement of the manuscript.

Abstract: The Abstract is well written and adequately presents the aim and the basic results of the study.

Introduction: The introduction section adequately covers the basic aim of the study.

-Authors could state the aim of the study as different paragraph at the end of Introduction.

Materials and Methods:  The materials and methods are adequately presented.

-Could authors add details about the questionaries used about the nutritional evaluation of the athletes?

-Did authors use calculation program for the total number of the participants? Please specify.

Results: The results of the study are analytically presented. Tables and Figures are adequate explain the findings of the study.

Discussion: The results of study are sufficiently discussed.

Conclusion: The conclusion is adequate and summarizes the main text.

Bibliography/References: The references used by the authors cover adequately the relative scientific field and the aims of the study.

Author Response

Point-by-Point Response to Reviewer’s Comments

We would like to sincerely thank to the reviewers for their helpful recommendations. We have seriously considered all the comments and carefully revised the manuscript accordingly. Revisions are highlighted in yellow through the manuscript in order to indicate where changes have taken place. We feel that the quality of the manuscript has been significantly improved with these modifications and improvements based on the reviewers’ suggestions and comments. We hope our revision will lead to an acceptance of our manuscript for publication in Nutrients.

In advance,

Kind regards

Reviewer 1

General comment: The research article entitled “Combined effects of citrulline plus nitrate-rich beetroot extract supplementation on maximal and endurance-strength and aerobic power in trained male triathletes: A randomized double-blind, placebo-controlled trial” is a well-organized study, with sufficient methodology and adequate description of the results. Some minor corrections are required for the improvement of the manuscript.

REVIEWER: Abstract: The Abstract is well written and adequately presents the aim and the basic results of the study.

AUTHORS: Thank you for your comment.

REVIEWER: Introduction: The introduction section adequately covers the basic aim of the study.

AUTHORS: Thank you for your comment.

REVIEWER: -Authors could state the aim of the study as different paragraph at the end of Introduction.

AUTHORS: Thank you for your recommendation. The authors have separated the aim of the study as different paragraph at the end of Introduction

REVIEWER: Materials and Methods:  The materials and methods are adequately presented.

AUTHORS: Thank you for your comment.

REVIEWER: -Could authors add details about the questionaries used about the nutritional evaluation of the athletes?

AUTHORS: Thank you for your interest. The authors have added this information in the participants section: To control that all athletes performed the planned diets, they performed 2 validated methods of dietary recall [42]. The first method was a food frequency questionnaire (FFQ) previously used in other studies with athletes [43,44] at the end of the study (T2). The second method was a seven-day dietary recall collected the week prior to T1 and during the week of T2. This method was used to check if the results of the FFQ were similar to those of this recall [45].

REVIEWER: -Did authors use calculation program for the total number of the participants? Please specify.

AUTHORS: Thank you for your interest. The sample was a convenience sample. The authors know this limitation and we have added the next sentence in the limitation section: “Moreover, sampling using a convenient, non-probabilistic sampling procedure may produce results that are not representative of the rest of the population. These limitations may underrepresent the results and may affect study outcomes.”

REVIEWER: Results: The results of the study are analytically presented. Tables and Figures are adequate explain the findings of the study.

AUTHORS: Thank you for your comment.

REVIEWER: Discussion: The results of study are sufficiently discussed.

AUTHORS: Thank you for your comment.

REVIEWER: Conclusion: The conclusion is adequate and summarizes the main text.

AUTHORS: Thank you for your comment.

REVIEWER: Bibliography/References: The references used by the authors cover adequately the relative scientific field and the aims of the study.

AUTHORS: Thank you for your comment.

Reviewer 2 Report

The manuscript entitled "Combined effects of citrulline plus nitrate-rich beetroot extract supplementation on maximal and endurance-strength and aerobic power in trained male triathletes: A randomized doubleblind, placebo-controlled trial" is an interesting one, the results are easy to follow, but I consider that it is not suitable for the Nutrients Journal.

However, if the publisher considers the manuscript suitable for this Journal, the authors should explain how they determined the concentrations used (if they only referred to other studies, I do not see the point of this study). If there is performance, it differs depending on the concentration. Highlight the novelty of the study (experiment). Statistical data analysis was not used.

Author Response

Point-by-Point Response to Reviewer’s Comments

We would like to sincerely thank to the reviewers for their helpful recommendations. We have seriously considered all the comments and carefully revised the manuscript accordingly. Revisions are highlighted in yellow through the manuscript in order to indicate where changes have taken place. We feel that the quality of the manuscript has been significantly improved with these modifications and improvements based on the reviewers’ suggestions and comments. We hope our revision will lead to an acceptance of our manuscript for publication in Nutrients.

In advance,

Kind regards

Reviewer 2

The manuscript entitled "Combined effects of citrulline plus nitrate-rich beetroot extract supplementation on maximal and endurance-strength and aerobic power in trained male triathletes: A randomized double-blind, placebo-controlled trial" is an interesting one, the results are easy to follow, but I consider that it is not suitable for the Nutrients Journal.

REVIEWER: However, if the publisher considers the manuscript suitable for this Journal, the authors should explain how they determined the concentrations used (if they only referred to other studies, I do not see the point of this study).

AUTHORS: Thank you for your interest. The doses of both supplements used in this study are based on effective doses reported in other studies when both supplements have been used individually. In this sense, the authors have explained this in the next paragraph: “Therefore, these 2 supplements (CIT and BEET) have been extensively studied individually showing improvements when are supplemented acutely (30-60 min before exercise) or in the short-term (7 days to 4 weeks) [33–35]. However, a possible combination of both for long-term (9 weeks), could further improve performance than either CIT or BEET separately. In this line, our research group presented that 10 weeks of creatine monohydrate and with β-hydroxy-β-methylbutyrate (HMB) oral co-supplementation showed a synergistic effect by increasing athletic performance, decreasing exercise-induced muscle damage and regulating hormonal behaviors in comparison to when they were taken in isolation [36,37].

REVIEWER: If there is performance, it differs depending on the concentration.

AUTHORS: Thank you very much for your interest. To the authors' knowledge, we have not found any studies showing dose-dependent effects of both supplements taken individually. However, based on the results obtained in the present study, the co-supplementation of citrulline and nitrate is more effective on performance than when taken individually.

REVIEWER: Highlight the novelty of the study (experiment).

AUTHORS: Thank you for your suggestion. The authors have highlighted the novelty of the study in the first paragraph of discussion section: “To the best of authors’ knowledge this is the first study that has evaluated the effect of the long-term (9 weeks) oral co-supplementation with 3 g/day CIT plus 2.1 g/day BEET (300 mg/day NIT) on maximal strength, endurance-strength performance test and aerobic power test in male triathletes. The main results indicated that this combination presented group-by-time on aerobic power measured by the Cooper Test, showing an increase in estimated VO2max in the mentioned test respect to PLG and CITG. On the other hand, although the maximal strength performance by HJUMP and DYN, and endurance-strength performance by 1-MAT, did not show significant differences in the group-time interaction, the CIT-BEETG showed a positive trend with respect to the other supplementation groups in the same tests. Moreover, only the CIT-BEETG displayed a significant increase on HJUMP and 1-MAT after 9 supplementation weeks.”

REVIEWER: Statistical data analysis was not used.

AUTHORS: Thank you very much for your comment. The main conclusion of the study is based on the results obtained using various statistical tests as indicated in the statistical analysis section. However, the sample was a convenient sample, and the authors know that is a limitation. This sentence has been added in the limitation section: “Moreover, sampling using a convenient, non-probabilistic sampling procedure may produce results that are not representative of the rest of the population. These limitations may underrepresent the results and may affect study outcomes.”

Round 2

Reviewer 2 Report

The authors responded to all my comments.
But it seems I don't know that they sent the manuscript to Nutrients, not Biology (

We would like to sincerely thank to the reviewers for their helpful recommendations. We have seriously considered all the comments and carefully revised the manuscript accordingly. Revisions are highlighted in yellow through the manuscript in order to indicate where changes have taken place. We feel that the quality of the manuscript has been significantly improved with these modifications and improvements based on the reviewers’ suggestions and comments. We hope our revision will lead to an acceptance of our manuscript for publication in Biology.)